# Robust Foundation Models Empowered RAN Intelligence for Reliable Embodied Robot Scenarios

## Abstract

The rapid development of large-scale AI has made intelligent robots increasingly viable for applications such as parcel sorting in warehouses. Coupled with advances in mobile communication, robots can now cooperate efficiently; however, conventional AI-based solutions often face low resource utilization and limited robustness, which hinders both sorting accuracy and handling efficiency. To address this, we propose a robust Foundation Model (FM)-empowered O-RAN framework that enables secure, robust, and real-time robot cooperation. An adaptive FM-splitting algorithm decomposes tasks into sequential sub-missions to improve sorting accuracy, while robustness training ensures resilience to environmental variations. Additionally, a cooperative path planning algorithm optimizes the number of active robots, reducing handling latency and energy consumption. Experiments demonstrate stable GPU utilization, up to 90% sorting accuracy, a 13.9% reduction in latency, and enhanced operational safety compared with conventional FM-based approaches.

## 1 Introduction

The rapid development of Artificial Intelligence (AI) is accelerating the intelligence of the Internet of Things (IoT) Xu et al. (2024), driving sixth-generation (6G) mobile communications to evolve from merely providing high-speed connectivity toward enabling intelligent and secure device interactions. Within this context, robots are emerging as key enablers in domains such as smart manufacturing, healthcare, and smart agriculture Kiyokawa et al. (2024). Owing to their cost-effectiveness, robots can collaborate to accomplish both repetitive and complex tasks, thereby reducing labor costs while ensuring operational reliability and safety.

However, several critical challenges still hinder the efficiency and safety. Firstly, it is challenging to achieve accurate parcel sorting due to limited onboard computing resources and the heterogeneous requirements of parcel distribution. Secondly, maintaining reliable and high-throughput information exchange is a significant challenge due to constrained communication resources. Thirdly, the considerable physical distance between robots and the cloud server introduces high transmission delays, which compromise low-latency communication.

We can deploy an Open Radio Access Network (O-RAN) at the network edge Tang et al. (2023); Polese et al. (2023) to optimize resource scheduling. Through RAN Intelligent Controllers (RIC), O-RAN enables customized communication resource scheduling to improve the accuracy and reliability of parcel sorting. The RIC consists of the Non-Real-Time RIC (Non-RT RIC) and the Near-Real-Time RIC (Near-RT RIC). The Non-RT RIC leverages rApps to support mission planning and assist robots in performing precise parcel classification. In contrast, the Near-RT RIC runs within the RAN to provide real-time parcel handling services with strict latency guarantees. Both RICs can dynamically allocate computing resources to generate safe and feasible motion paths, thereby enhancing not only the efficiency and accuracy of parcel handling in multi-robot cooperation.

Unfortunately, existing AI algorithms remain highly dependent on specific datasets. As one of the most promising AI paradigms, the Foundation Model (FM) leverages a transformer architecture with self-attention to estimate RAN resources with high precision Sun et al. (2024). Compared with traditional ensemble learning methods Stenhammar et al. (2024), FM enables broader task

generalization through self-supervised learning. By training on large-scale datasets, FM can support O-RAN in synthesizing new robotic scenario data, thereby enhancing the RIC's capability to make precise scenario estimations for efficient and reliable parcel sorting and handling. Furthermore, FM allows the RIC to explore customized RAN resource scheduling strategies for seamless and collision-free robot cooperation. In return, O-RAN provides real-time scheduling feedback to FM, enabling accurate and robust model adaptation to diverse distribution service requirements.

With this motivation, we propose a robust FM-empowered O-RAN framework. Unlike the conventional O-RAN architecture Tang et al. (2023), our approach introduces a hierarchical FM training paradigm that enables multi-component cooperation. The main contributions of this work are summarized as follows.

- We propose a robust FM-empowered O-RAN framework, which represents the first exploration of integrating Foundation Models with O-RAN in robot-oriented scenarios. Within O-RAN, we construct customized rApps and xApps in the respective RICs to enhance RAN optimization through fine-grained communication resource scheduling. Based on these scheduling results, the framework can coordinate an appropriate number of robots to achieve reliable and safe parcel sorting and handling. Furthermore, the terminal–edge cooperative paradigm improves computing resource utilization and shortens FM training time.

- To enhance sorting accuracy, we design an adaptive FM-splitting algorithm deployed in the Non-Real-Time RIC (Non-RT RIC). This algorithm dynamically partitions the FM to generate customized rApps tailored to specific parcel distribution areas, destinations, and latency requirements. These rApps guide robots to perform cooperative parcel sorting in a pipelined manner, thereby ensuring high-accuracy sorting.

- To achieve low-latency and reliable parcel handling, we develop a cooperative path planning algorithm implemented in the Near-Real-Time RIC (Near-RT RIC). This algorithm enables xApps to determine the optimal number of robots for cooperative handling based on their positions, velocities, and energy levels. It then optimizes handling paths to improve planning efficiency while reducing energy consumption, thereby ensuring timely and highly reliable parcel handling.

## 2 ROBUST FOUNDATION MODEL EMPOWERED RAN FRAMEWORK

In this section, we provide detailed descriptions with three functional layers: (i) Robust cooperative FM construction, (ii) parcel sorting with Non-RT rApps, and (iii) parcel handling with Near-RT xApps.

As shown in Fig. 1, the FM construction is regarded as the foundation of the framework with two core components: information collection and FM model aggregation. The former allows robots to implement cooperative data collection using embedded sensors, including the number of parcels, the status of neighbors, and the surrounding physical obstacles for lightweight sub-FMs requisitions. With all the sub-FMs, the edge RAN can aggregate the sub-FM to an FM to implement global robot management and RAN resource optimization for accurate parcel sorting and real-time handling.

We implement the parcel sorting on the SMO side with a time-insensitive characteristic for accurate sorting. We mainly analyze the sorting mission from the perspective of different delivery destinations. Explicitly, we propose an adaptive FM splitting algorithm to implement accurate parcel sorting based on the different delivery destinations. Specifically, our algorithm can dynamically split the FM into three customized sub-FMs based on parcel information in the Non-RT RIC (at the SMO): Region-FM-rApp (R-FM-rApp), Destination-FM-rApp (D-FM-rApp), and Latency-FM-rApp (L-FM-rApp). The R-FM-rApp classifies parcels according to distribution regions.

We then propose an adaptive model splitting algorithm to acquire customized xApps, the Energy-FM-xApp (E-FM-xApp) and the Path-Planning-FM-xApp (PP-FM-xApp), based on robot information such as positions, velocities, and energy for low-latency handling operations.

Based on the E-FM-xApp, we can enable robots to estimate changes of energy for associating with the feasible parcel handling mission. The estimation result conducts robots to invite suitable collaborators to move to suitable positions in advance using an information request way with an event-

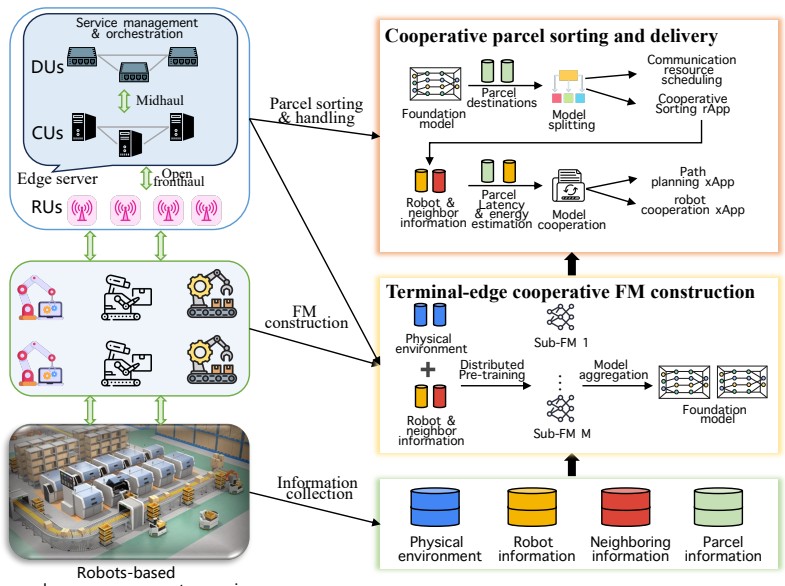

Figure 1: Illustration of robust FM-empowered O-RAN framework for parcel sorting and handling in robot-based warehouse management scenarios.

triggered mechanism for cooperative and reliable parcel handling Wang et al. (2023). We also enhance the information exchange efficiency by only transmitting lightweight E-FM-xApp parameters. This makes use of achieving sustainable parcel handling performance with enhanced robot cooperation.

The PP-FM-xApp, running on the CU side, enables robots to implement dynamic path optimization with the prediction results of the trajectories of neighboring robots based on their positions and parcel information. The handling Path optimization is estimated with two key parts: autonomous path adjustment and cooperative path planning. Explicitly, robots first engage in decision-sharing behavior to obtain handling path decisions from their neighbors. We can detect whether overlapping parcel handling paths exist or not. When overlapping parcel handling paths are detected, the robots can utilize the PP-FM-xApp to generate a new path for autonomous path adjustment. We can invoke a robot negotiation algorithm to implement cooperative path adjustment Cao et al. (2024). This method can jointly consider potential physical collisions and parcel handling latency to optimize newly generated paths for real-time parcel handling.

# 3 PARCEL SORTING AND HANDLING ALGORITHM DESIGNS WITH ROBUST FMS

In this section, we provide corresponding algorithms for accurate parcel sorting and handling, which consist of three parts: *Robust FM construction FM splitting for accurate parcel sorting* and *FM cooperation for real-time parcel handling*.

## 3.1 ROBUST FM CONSTRUCTION

Foundation models are vulnerable to several security risks during training, including data poisoning (injection of malicious samples into large-scale corpora), privacy leakage, and backdoor insertion (hidden triggers that cause targeted misbehavior). Moreover, these models often exhibit high sensitivity to adversarial perturbations, which compromises their reliability in safety-critical applications. To mitigate these risks, we design a robust training method: adversarial training formulates learning

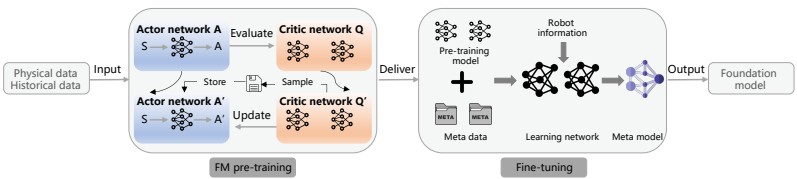

Figure 2: Illustration of cooperative FM construction.

as a min–max optimization problem Razaviyayn et al. (2020):

$$\min_{\theta} \; \mathbb{E}_{(x,y)} \left[ \max_{\delta \in \mathcal{S}} L\big(f_\theta(x + \delta), y\big) \right],\tag{1}$$

where $\theta$ is model parameters ( weights of a neural network); $(x, y)$ is the input sample $x$ with its ground-truth label $y$; $f_\theta(x)$ is model prediction for input $x$ under parameters $\theta$l $L(\cdot, \cdot)$ is cross-entropy measuring prediction error; $\delta$ is adversarial perturbation applied to input $x$.; $\mathcal{S}$ is a feasible perturbation set; $\mathbb{E}_{(x,y)}[\cdot]$ is expectation over the training data distribution. We explicitly enhance resistance against input perturbations. Differential privacy can be incorporated to limit information leakage from individual samples. The FM acquisition is illustrated in Fig. 2. The pre-training process is quantified as a Stochastic Game (SG) problem with a tuple $\{S_i, A_i, \mathcal{T}, r_i\}$, where $\mathcal{T}$ is a transfer function and $r_i$ is a reward function. The $R_i$ is formulated by

$$r_i(S_i, A_i) = \frac{1}{k_i} \sum_{k=1}^{k_i} [\Delta D_{j,k} + \Delta D_{i,k}],\tag{2}$$

where $k_i$ is the number of parcels; $\Delta[f] = f(t-1) - f(t)$; $\alpha_i$ and $\beta_i$ can be set as 5 and 0.05, respectively. The robots can learn a feasible pre-training FM model from the critic network by minimizing the reward:

$$\text{FM}_{\text{pre}}(\theta^\mu) = E_{S,A \backsim \Omega} \left[ \frac{1}{M} \sum_i \nabla_{\theta^\mu} \mu(A_i | S_i) \nabla_{A_i} Q^\mu(S_i, (A_1, \cdots, A_M)) |_{A_i = \mu(S_i)} \right].\tag{3}$$

Based on the pre-training FM model, A data sampling operation can be enabled to decompose the data for customized parcel sorting and handling:

$$\mathcal{D}_{\mathcal{K}}^{\text{sorting}} = \{(x_i, c_i, y_i)\}_{i=1}^K, \quad \mathcal{D}_{\mathcal{K}}^{\text{handling}} = \{(x_j, c_j, y_j)\}_{j=1}^M,\tag{4}$$

where $(x_i, c_i, y_i)$ and $(x_j, c_j, y_j)$ are information vectors to reflect the parcel sorting and handling, respectively. We can obtain an accurate parcel sorting FM:

$$\text{FM}_{\text{sorting}} = -\sum_{c=1}^C \log p_\theta(c \mid x, c),\tag{5}$$

where $x$ is the input information; $c$ is the training label; $p$ is the prediction function. The parcel handling FM is formulated by

$$\text{FM}_{\text{handling}} = \|\mathbf{p} - \hat{\mathbf{p}}\|_1 + \lambda_R \, d_{\text{SO(3)}}(\mathbf{q}, \hat{\mathbf{q}}),\tag{6}$$

where $p$ and $\hat{\mathbf{p}}$ are predictive and actual translation actions of robots, respectively; $\mathbf{q}$ and $\hat{\mathbf{q}}$ are predictive and actual orientation actions of robots, respectively.

## 3.2 FM SPLITTING FOR ACCURATE PARCEL SORTING

We present the implementation process of parcel sorting illustrated in Fig. 3. We propose an adaptive model splitting algorithm that utilizes an attention mechanism to acquire multiple sub-FMs. Specifically, we first formulate the feature vector $h_k$ for accurate parcel sorting with different delivery regions, destinations, and latency:

$$h_k = E_\theta(x_k, \text{Env}),\tag{7}$$

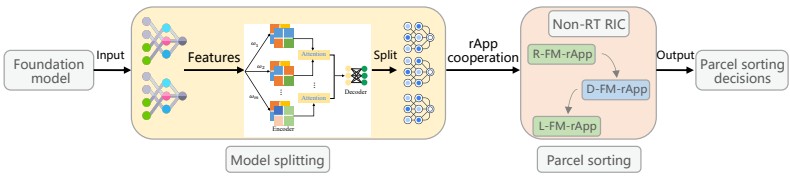

Figure 3: Illustration of parcel sorting.

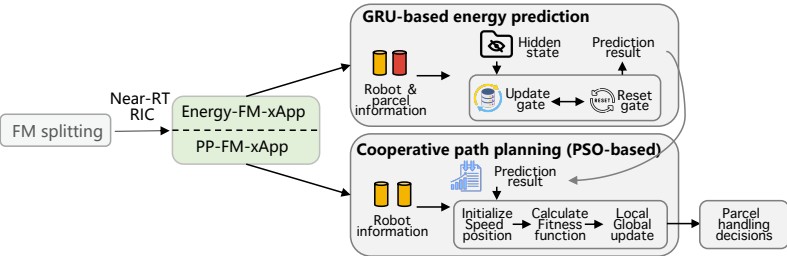

Figure 4: Illustration of cooperative parcel handling with two kinds of xApps, energy-FM-xApp (E-FM-xApp) and Path Planning (PP-FM-xApp).

where $E_\theta$ is the shared encoder network parameterized by $\theta$. It maps the input pair $(x_k, \mathrm{Env})$ into a latent representation (embedding); $x_k$ is the raw input features of parcel $k$; $\mathrm{Env}$ is the environment information. The corresponding attention heads are then formulated as

$$p^{\mathrm{R}} = \mathrm{softmax}(W_{\mathrm{R}}h_k), \quad p^{\mathrm{D}} = \mathrm{softmax}(W_{\mathrm{D}}h_k), \quad p^{\mathrm{L}} = \mathrm{softmax}(W_{\mathrm{L}}h_k), \tag{8}$$

where $W_{\mathrm{R}}$, $W_{\mathrm{D}}$, and $W_{\mathrm{L}}$ are weight matrices. We employ the decoder to connect these attention heads with model parameters using a connection probability:

$$\mathcal{L}_{\mathrm{R-FM-rApp}} = \alpha\big(-\log p_{y^{\mathrm{R}}}^{\mathrm{R}}\big), \quad \mathcal{L}_{\mathrm{D-FM-rApp}} = \beta\big(-\log p_{y^{\mathrm{D}}}^{\mathrm{D}}\big), \quad \mathcal{L}_{\mathrm{L-FM-rApp}} = \gamma \sum_k C_{y^{\mathrm{L}},k}\, p_k^{\mathrm{L}}, \tag{9}$$

where $\alpha$, $\beta$, and $\gamma$ are weights controlling the contribution of the region classification loss. $p^{\mathrm{R}}$, $p^{\mathrm{D}}$, and $p_k^{\mathrm{L}}$ are probability distributions over all region classes; $y^{\mathrm{R}}$ and $y^{\mathrm{D}}$ are ground-truth class labels for the current samples. $C_{y^{\mathrm{L}},k}$ is the penalty of predicting class $k$ when the true class is $y^{\mathrm{L}}$.

### 3.3 MODEL COOPERATION FOR REAL-TIME PARCEL HANDLING

We propose a cooperative parcel handling algorithm based on information on robots and parcels with two main implementation steps shown in Fig. 4: Gated Recurrent Unit (GRU)-based energy prediction and Particle Swarm Optimization (PSO)-based cooperative path planning.

In the Near-RT RIC, we input the robot and parcel information into the input gate. We then activate the reset gate $r_t$ using the $\sigma$ function to determine how much of the previous hidden state should be forgotten when calculating the new hidden state:

$$r_t = \sigma(W_r x_{\mathrm{en},t} + U_r h_{t-1} + b_r), \tag{10}$$

where $W_r$, $x_{\mathrm{en},t}$, $U_r$, $h_{t-1}$ are weight matrix, input feature vector at time $t$, weight matrix for hidden state $h_t$, and the previous hidden state, respectively. $b_r$ is the bias vector. We can train the GRU network to acquire an accurate energy prediction result:

$$\mathcal{L}_{\mathrm{pre}} = \frac{1}{T} \sum_{t=1}^{T} \big(\hat{E}_t - E_t\big)^2, \tag{11}$$

where $\hat{E}_t = W_y h_t + b_y$. With the prediction results, for robot $i$ at the dimension $d$, the position and velocity are updated as

$$V_{i,d} = \epsilon V_{i,d} + a_1 r_1 (X_d^{\mathrm{g}} - X_{i,d}) + a_2 r_2 (X_d^{\mathrm{p}} - X_{i,d}), \tag{12}$$

$$X_{i,d} = X_{i,d} + V_{i,d}, \tag{13}$$

where $\epsilon$ is the inertia parameter, $a_1$ and $a_2$ are acceleration coefficients; $r_1$ and $r_2$ are constrained in $[0, 1]$; $X^{\mathrm{g}}$ and $X^{\mathrm{p}}$ are global and local optimization results that are updated by

$$X_d^{\mathrm{g}} = \arg \min_{X_d} g(X_{i,d}) = \sum_{j=1}^{M} \left\| X_{i,d} - h_j \right\|_2, \quad X_d^{\mathrm{p}} = \arg \min_{X_d} g^{'}(X_{i,d}) = \sum_{j=1}^{M} \left\| X_{i,d} - h_j^{'} \right\|_2, \tag{14}$$

where $h_j$ and $h_j^{'}$ are global and local prediction values, respectively.

## 4 PERFORMANCE EVALUATION

**Scenario Design**: We construct a 3 km $\times$ 3 km robot-based parcel sorting and handling scenario under varying numbers of robots and parcels with different weights using Simio, a digital twin simulation software, on the NVIDIA GB200 NVL72 server. The parcels are deployed randomly in the virtual scenario. We equip heterogeneous onboard sensors, such as cameras, LiDAR, and the Inertial Measurement Unit (IMU), on the robots to enable environmental data collection. The moving collection path is planned using the differential drive kinematics theory Wei et al. (2024), combining the information of the Global Positioning System (GPS) sensor. The parcel handling paths are estimated through energy consumption, which is detected by voltage sensors.

**Robust FM Training Deployment**: We equip the Manifold, an embedded computer, on the robots to provide computing resources for cooperative FM training based on local information. We use the popular Pytorch framework to construct the DDPG algorithm for FM training with actor and critic networks Chen et al. (2021). Each network consists of three hidden layers, and each hidden layer has 64 neural units. The learning rate and batch size are set to 0.99 and 128, respectively. We use a sigmoid activation function to implement data training. Considering the diversity of parcels, we adopt a stochastic gradient descent optimization method to explore feasible sorting and handling decisions through an iterative learning process. We can implement further FM training and optimization based on the existing dataset from the intelligent robots for warehouse management dataset at the edge RAN. Based on this, we implement the fine-tuning training based on the parcel sorting and handling dataset using a Euclidean distance based meta-learning method. We also deploy a conventional GRU network with the Adam optimization method Ławryńczuk & Zarzycki (2025).

**Edge RAN Deployment** For the edge RAN design, we use the Bubble RAN solution to replicate the AI-RAN solution with the MX-PDK development. In the virtual RAN environment, the robot interaction dataset is used to achieve robot cooperation for parcel handling with key interaction variables: robot ID, type of parcels, handling start and end times, robot communication types, and data packet size. The robots can connect to the Bubble RAN via the cellular network. The testbed is shown in Fig. 6. Several key metrics are selected to highlight our contributions.

1. *FM training loss*: We leverage the metric to evaluate the FM training performance in the given robots-based parcel sorting and handling scenario.
2. *GPU utilization*: We use the metric to reflect the stability of cooperative training.
3. *Parcel sorting accuracy*: This metric reflects the accuracy of FM training and FM splitting for accurate sorting.
4. *Path planning efficiency*: This metric estimates the performance of robot cooperation by analyzing sorting results with varying numbers of robots and parcels.
5. *System response time*: We leverage the metric to evaluate the robot management efficiency.

We provide three state-of-the-art benchmarks for comparison: a centralized FM implementation method He et al. (2024) based on a cloud computing pattern, a distributed FM implementation method Chen et al. (2024) with cooperative FM construction among robots, and a GRU-based edge RAN implementation method Ławryńczuk & Zarzycki (2025). The experiment parameters are summarized in Tab. 1.

In Tab 2, we use different $\epsilon$, namely, an upper bound on the adversarial perturbation power, to test the adversarial robustness performance. These results demonstrate that min–max adversarial

Table 1: Experiment parameters.

| Parameter description | Value |
|---|---|
| Parcel count | [30, 70] |
| Average weight of parcels | [8 kg, 16 kg] |
| Parcel count w/ destination changes | [6, 14] |
| Robot count | [20, 60] |
| Average moving velocity of the robots | 8 km/h |
| Energy budget of a robot | $1.4 \times 10^2$ kJ |
| Energy consumption | [51, 89] kJ/km |
| Delivery area | 3000 m $\times$ 3000 m |
| Learning rate | [0.001, 0.009] |
| Transmission power of robots | [60 mW, 100 mW] |
| Communication bandwidth | [50 MHz, 100 MHz] |
| Gaussian White Noise | -96 dBm/Hz |
| Acceptable maximal system latency | 15 minutes |

Table 2: Adversarial robustness performance

| Method | Clean | $\epsilon$=1/255 | 2/255 | 4/255 | 8/255 | ECE↓ |
|---|---|---|---|---|---|---|
| FM-Based (no defense) | 84.7 | 62.5 | 41.2 | 18.9 | 5.3 | 4.8 |
| AdvTrain (PGD-10) Qi et al. (2024) | 82.9 | 74.6 | 66.3 | 52.1 | 28.7 | 3.2 |
| Ours | 81.4 | **76.8** | **69.1** | **55.9** | **31.5** | **2.6** |
| AdvTrain + DP-SGD Thakkar et al. (2024) | 79.6 | 71.1 | 63.0 | 48.2 | 26.9 | 3.0 |

training frameworks can substantially enhance the robustness of foundation models against norm-bounded perturbations while maintaining competitive clean accuracy. Table 3 presents the clean accuracy, backdoor attack success rate (ASR), and membership inference attack (MIA) AUC of different defense strategies. These results highlight that no single defense dominates across all dimensions: adversarially-informed defenses are most effective against backdoors, while differential privacy provides stronger resistance to privacy leakage. Thus, our solution combines complementary approaches to yield a more balanced security-robustness trade-off for foundation model training.

Fig. 5(a) illustrates the training loss of the cooperative FM under different numbers of robots, ranging from 20 to 60. We see that across all robots' numbers, the training loss decreases rapidly before iteration 200. The performance reflects efficient convergence at the initial stage. As training progresses, the loss gradually flattens and stabilizes, indicating that the model has reached convergence. In addition, we observe that configurations with more robots achieve lower training losses compared to scenarios with fewer robots. This trend demonstrates that increasing the number of cooperating robots enhances the representational capacity and leads to improved convergence performance. The results imply that our solution can achieve satisfactory FM acquisition for efficient parcel sorting and handling.

Fig. 5(b) depicts the GPU utilization of the cooperative FM framework under varying numbers of robots and parcels, respectively. The GPU utilization is compared across scenarios with 20, 30, and 40 robots. The results indicate that increasing the number of robots generally leads to higher and more stable GPU utilization. This suggests that larger cooperative groups are able to better exploit computational resources to minimize idle time and ensure efficient parallel implementation with low execution latency.

Fig. 5(c) illustrates the confidence error of FM training with respect to the number of iterations. The confidence error quantifies the gap between the predicted probability distribution and the ground truth, thereby reflecting the reliability of the decision-making. We see that our solution consistently achieves lower confidence errors compared to other benchmarks. This demonstrates its ability to converge faster and maintain more reliable training predictions. Overall, the results highlight that

Table 3: Security evaluation performance

| Defense | Clean Acc (%) | ASR↓ | MIA AUC↓ | Notes |
|---|---|---|---|---|
| None (backdoored) | 84.2 | 92.7 | 0.86 | poisoned fine-tune |
| Fine-tune (clean data) | 83.9 | 38.4 | 0.79 | 5 epochs |
| Spectral Sign. (pruning) | 83.1 | 21.6 | 0.77 | top-1% removal |
| Activation Clustering | 82.7 | 17.9 | 0.76 | $k=2$ clusters |
| Gradient Shaping + AdvTrain | 82.1 | **8.3** | 0.74 | joint defense |
| DP-SGD ($\varepsilon=4$) | 80.5 | 12.7 | **0.63** | $\sigma=1.2$, clip=1.0 |

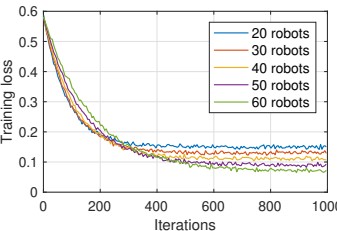

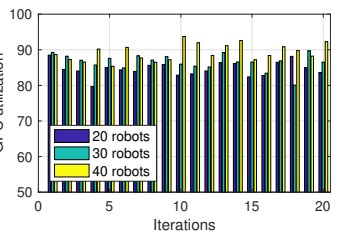

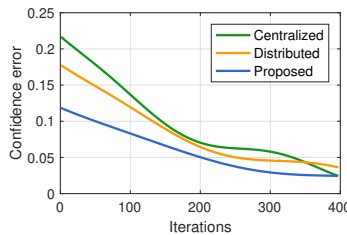

(a) Training loss of cooperative FM with different numbers of robots.

(b) GPU utilization with different numbers of robots.

(c) Confidence error of FM training vs. iterations.

Figure 5: Performance evaluation of FM training.

our solution enhances both the convergence speed and the robustness of FM training, leading to safer and more reliable robot cooperation in parcel sorting and handling.

We compare parcel sorting accuracy in Fig. 7(a) under different numbers and types of parcels. With the random deployment of 40 robots with an average weight of 12 kg, our solution achieves the highest sorting accuracy compared to both benchmarks. This is due to our FM assisting O-RAN in scheduling communication resources to ensure seamless cooperation among robots for efficient parcel sorting. Furthermore, our solution decouples missions into three sorting steps using our model-splitting method, which guarantees accurate parcel sorting through streamlined pipeline operations. The latency-FM-rApp also enables robots to sort parcels in real time for low-latency requirements. Our solution improves the sorting accuracy by 7.1%, 13.8%, and 15% compared to centralized, distributed, and GRU-based solutions, respectively.

With the sorting results, we evaluate the efficiency of path planning, the ratio of the optimal path to the actual path with varying numbers of parcels, as illustrated in Fig. 7(b). Under the same deployment conditions as those in Fig. 7(a), our solution consistently achieves a high path planning efficiency of up to 90% compared to both benchmarks. This is because our energy-FM-xApp allows robots to select a feasible number of collaborators by estimating the energy consumption for cooperative parcel handling. Based on these estimation results, the PP-FM-xApp collaborates with the robots to implement cooperative path planning through information exchanges utilizing available communication resources. Our solution shortens the handling paths by 13.9% and 28.2% compared to distributed and centralized solutions, respectively.

Finally, we compare system implementation latency under different numbers of parcels shown in Fig. 7(c). With the same deployment scheme as shown in Fig. 7(a), we find that our solution consistently achieves the lowest implementation latency compared to both benchmarks. This improvement is attributed to our DDPG algorithm, which reduces the FM implementation time through a cooperative training manner by exchanging implementation actions. Additionally, our solution enables the O-RAN to collaborate with the computing resources of different robots to further accelerate FM training. We can dynamically schedule O-RAN communication resources to ensure reliable information exchanges for cooperative parcel sorting and handling. Our solution reduces system

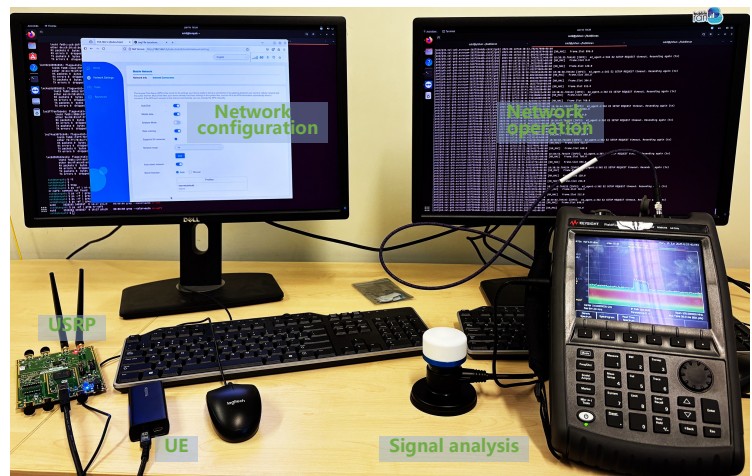

Figure 6: Illustration of edge RAN configuration.

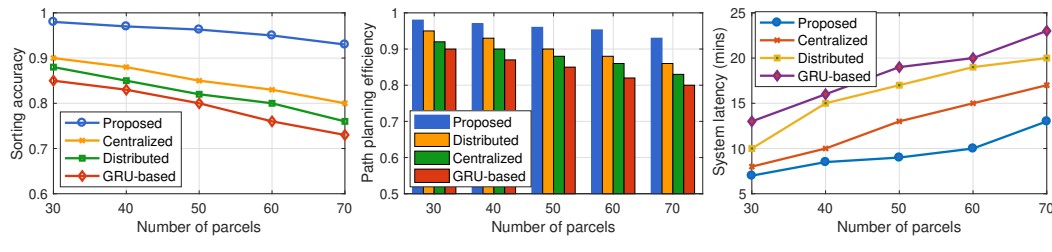

(a) Parcel sorting accuracy vs. number of parcels.

(b) Path planning efficiency vs. number of parcels.

(c) System latency vs. number of parcels.

Figure 7: Performance evaluation under different numbers of robots.

implementation latency by 15.8% and 30.4% compared to the distributed and centralized solutions, respectively.

## 5 CONCLUSION

In this paper, we have proposed a robust and safe FM-empowered RAN framework to assist robots in achieving accurate and real-time parcel sorting and handling in a mobile robot scenario. We first designed an FM splitting algorithm to acquire customized sub-FMs for accurate parcel sorting and real-time parcel handling. These sub-FMs can empower rApps to provide effective parcel sorting decisions based on different delivery regions, destinations, and latency. The sub-FMs can also assist robots in planning feasible handling paths by designing corresponding energy consumption xApps and path planning xApps. The simulation results demonstrated that our solution achieves accurate parcel sorting and low-latency parcel handling under different scenarios with various numbers of parcels and robots.

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

# A    APPENDIX

## A.1    RELATED WORK

We provide the state-of-the-art investigations and discussions for foundation model and AI-RAN in robust robot scenarios.

**Foundation Model for robots-based scenarios**: To ensure accurate robot navigation, the authors in Xia et al. (2024) proposed an agent-based foundation model as the new training paradigm for effective environmental sensing and robot navigation. The solution can generate cross-domain actions consistent with sensing information, paving the way to realize interactive and collaborative robots. However, the FM training can lead to the issue of security due to frequent information transmission between robots and edge RAN. The authors in Chen et al. (2025) developed a FASTNav to train a lightweight LLM, for reliable robot navigation by reduce the transmission data sizes. The proposed method contains three modules: fine-tuning, teacher-student iteration, and language-based multi-point robot navigation. The experiment results the solution can perform high secure and low latency robot navigation. To ensure reliable path planning, the authors in Qi et al. (2025) proposed a 3-step trajectory optimization framework for generating a jump motion for a humanoid robot. The framework can joint detect robot postures, centroidal angular momentum, and landing foot placement for reliable path planning. Nonetheless, the existing works neglect the communication resource constraint with high-frequency information transmission which can also cause lowly robust robot cooperation in 6G application scenarios.

**RAN empowered reliable communication for robot cooperation**: The reliable robot cooperation needs the support of suitable communication resources. It is important to schedule communication resources effectively by RAN. The authors in Baruffa et al. (2024) developed a testbed architecture that combines contemporary communication and cloud technologies to provide microservice-based mobile applications with the ability to offload part of their tasks to cloud/edge data centers connected by multi-RAT cellular networks. In addition, the authors in Bolla et al. (2023) presented an optimization problem to further explore redundant radio bearers for each robot. Such problem extends the

current specification on redundant transmissions. Several heuristic methods have been developed to meet the time-scale requirements that cannot be achieved through exhaustive search. However, such methods might still lead to high communication costs due to high-frequency information exchanges among robots. The authors in Chinchilla-Romero et al. (2024) proposed a centralized control solution for automated mobile robots. A techno-economic analysis was designed to assess the total system cost in an Industry 4.0 robot environment. A sensitivity analysis was also included for the solution identifying the variables with great impact on the system cost. Unfortunately, the existing works ignore the undetermined numbers of robots and missions, which can expose significant pressures for real-time computing and reliable communications among robots.

Based on the discussions mentioned-above, the existing works mainly focus on the study of RAN optimization and FM training while few of investigations pay attention to variability of numbers of robots and missions. In addition, it is difficult to ensure reliable and robust FM training based on the limited computing resources of edge RAN. On the other hand, the existing work may also lead to low-efficiency robot cooperation due to the neglect of change in physical environments. In this case, a terminal-edge cooperative robust FM training is feasible to optimize RAN resources for reliable, accurate, and low-latency robot services.

### A.2 PROBLEM FORMULATION

We formulate corresponding objective function to optimize the robots-based parcel sorting and handling with given constraints.

#### A.2.1 ANALYSIS OF DATA COLLECTION

The robots can implement cooperative sensing by dynamically changing self-positions. We first formulate the sensing feasibility constraints based on robots' positions and data size:

$$x_{ist} \leq \sum_{v \in \mathcal{N}(s)} m_{ivt}, \tag{15}$$

$$z_{ist} \leq \bar{q}_s \, x_{ist}, \tag{16}$$

where $s \in \mathcal{S}$ is sensing site with site set $\mathcal{S}$; $v \in \mathcal{V}$ is the spatial position of robots with position set $\mathcal{V}$; $x_{ist} \in \{0, 1\}$ and $m_{ivt} \in \{0, 1\}$ are sensing decision indicator and position indicator, respectively. $x_{ist} = 1$ when robot $i$ implements the sensing operation for site $s$ at time $t$; $m_{ist} = 1$ denotes robot $i$ is in the position $v$ at time $t$; $z_{ist}$ is a continued variance to present the data size (bits) that robot $i$ collects at site $s$; $\bar{q}_s$ is the maximal data sizes that site $s$ can provide. The two constraints can assist robots in optimizing self-positions for cooperative sensing.

In addition, we need to ensure comprehensive sensing by improving sensing coverage:

$$\sum_{i \in \mathcal{M}} \sum_{t \in \mathcal{T}} z_{ist} \geq Q_s, \tag{17}$$

where $Q_s$ is the minimal acceptable sensing coverage.

#### A.2.2 ANALYSIS OF PARCEL SORTING ACCURACY

For the parcel sorting accuracy, we give detailed discussions considering delivery regions, destinations, and latency. Explicitly, in terms of the delivery region, we enable the R-FM-rApp to estimate the current sorting result is whether in the given delivery regions or not:

$$R_k = \mathbb{I}[\, \hat{z}_k \in \mathcal{Z}_k^{\mathrm{srv}} \,], \tag{18}$$

where $R_k \in \{0, 1\}$ is an indicator; $\hat{z}_k$ is the estimation result; $\mathbb{I}$ is an indicator function. Based on this, we can give the sorting accuracy $A_{\mathrm{Ra}}$ considering the delivery regions as follows:

$$A_{\mathrm{Ra}} = \frac{\sum_{k \in \mathcal{K}} w_k \, R_k}{\sum_{k \in \mathcal{K}} w_k} \geq A_{\mathrm{Ra,min}}, \tag{19}$$

where $w_k$ is a weight of parcel $k$ for different priorities; $A_{\mathrm{Ra,min}}$ is the acceptable minimal sorting accuracy.

Considering the delivery destinations, we formulate the corresponding accuracy model. Before that, we first formulate the destination accuracy indicator $D_k$:

$$D_k = \mathbb{I}[\, \hat{z}_k = g_k \,], \tag{20}$$

where $g_k$ is the real delivery destination of parcel $k$. Based on this, we can formulate the destination accuracy model by

$$A_{\text{dest}} = \frac{\sum_{k \in \mathcal{K}} w_k \, D_k}{\sum_{k \in \mathcal{K}} w_k}. \tag{21}$$

With the sorting accuracy for different delivery regions, the sorting accuracy for different destinations $A_{\text{dest}|\text{Ra}}$ is formulated by

$$A_{\text{dest}|\text{Ra}} = \frac{\sum_k w_k \, D_k \, R_k}{\sum_k w_k \, R_k + \varepsilon}, \tag{22}$$

where $\varepsilon > 0$ is a constant value.

In terms of delivery latency, we can further sort the parcels for accurate and real-time handling. Similarly, we first give the latency accuracy indicator $S_k$:

$$S_k = \mathbb{I}[\, t_k \leq d_k \,], \tag{23}$$

where $t_k$ and $d_k$ are practical parcel sorting time and maximal acceptable parcel sorting time, respectively. Then, we give the joint optimization with accurate destination and required latency as follows:

$$A_{\text{SLA}} = \frac{\sum_{k \in \mathcal{K}} w_k \, D_k \, S_k}{\sum_{k \in \mathcal{K}} w_k}, \tag{24}$$

where $A_{\text{SLA}}$ is the accuracy indicator for joint optimization of delivery destination and latency. In this case, we further give the whole sorting accuracy $A_{\text{mix}}$:

$$A_{\text{mix}} = \lambda_1 A_{\text{range}} + \lambda_2 A_{\text{dest}} + \lambda_3 A_{\text{SLA}} \geq A_{\text{mix,min}}, \tag{25}$$

where $\lambda_i \geq 0$, $\sum_i \lambda_i = 1$; $A_{\text{mix,min}}$ is the acceptable minimal whole sorting accuracy.

### A.2.3 ANALYSIS OF PARCEL HANDLING LATENCY

We analyze the parcel handling latency with two parts: cooperative path planning and dynamic path adjustment. For the cooperative path planning, the latency mainly includes service latency $\tau^{\text{svc}}$ (namely the time of loading and unloading parcels), handling latency, and the waiting latency for giving way to other robots with collision avoidance. Specifically, the service latency for robot $i$, $\tau_i^{\text{svc}} > 0$, is regarded as a constant value with static loading and unloading time. Regarding the handling latency $\tau_i^{\text{hand}}$ of robot $i$, the planned path for robot $i$ can be represented as $\text{PP}_i$. The average moving velocity of robot $i$ is $v_i$. We can calculate the handling latency as

$$\tau_i^{\text{hand}} = \frac{\text{PP}_i}{v_i}. \tag{26}$$

The waiting latency is defined to avoid physical collisions based on planned paths. In this case, we can use a graph $G(V, E)$ to present the cooperative parcel handling, where $V$ is the all robot node set and $E$ is the handling path set. For robot $i$, we firstly need to guarantee $\sum_{i \in \mathcal{M}} m_{ivt} \leq 1$, namely each node matches one robot at the same time $t$. In addition, the edge capacity constraint can be formulated as

$$\sum_{i \in \mathcal{M}} \left( a_{iuv,t} + a_{ivu,t} \right) \leq \text{cap}_{uv}, \tag{27}$$

where $a_{iuv,t} = 1$ denotes robot $i$ moves from node $u$ to $v$, and verse visa. $\text{cap}_{uv}$ is the maximal robot capability at the edge $(u, v)$. In this case, when robot $i$ and $j$ move towards the same edge at the same time $t$, the waiting latency $W_{i,t}$ can be formulated by

$$W_i = \left( \sum_{i \neq j} \mathbb{I}[a_{i,j,t'} = 1] \cdot \Delta_{i,j,t'} \right), \tag{28}$$

where $\mathbb{I}[a_{i,j,t'} = 1$ is an indicator function to present there exist a conflict between robot $i$ and $j$; $\Delta_{i,j,t'}$ is the waiting time of robot $i$. In this context, we can constrain the whole path planning latency of robot $i$ by

$$t_i^{\mathrm{PP}} = \tau_i^{\mathrm{hand}} + \tau^{\mathrm{svc}} + W_i. \tag{29}$$

The robots can re-adjust their moving paths to cope with the time-varying parcel handling scenarios. We analyze the dynamic path adjustment with two parts: the path adjustment latency $W_{\mathrm{adj},i}$ and the revised handling completion latency $C_{\mathrm{adj},t}$. The former can be formulated as

$$W_{\mathrm{adj},i} = \sum_{t' \leq t} \mathbb{I}[a_{i,v,t'} = 0] \cdot \mathrm{Id}(t'), \tag{30}$$

where $\mathbb{I}[a_{r,v,t'} = 0]$ is an indicator function, where $a_{r,v,t'} = 0$ denotes robot $i$ is in the idle state in node $v$ at time $t$. $\mathrm{Id}(t')$ denotes the idle time of robot $i$. The latter can be represented as

$$C_{\mathrm{adj}}(i) = C_i^u + \tau_{\mathrm{rep}}, \tag{31}$$

where $C_i^u$ is the current handling latency based on the previous path planning decision; $\tau_{\mathrm{rep}}$ is the computing latency for path adjustment. In this context, the whole parcel handling latency $t_i^{\mathrm{handle}}$ is constrained by

$$t_i^{\mathrm{handle}} = W_{\mathrm{adj},i} + C_{\mathrm{adj}}(i) + t_i^{\mathrm{PP}} \leq t_{i,\max}^{\mathrm{handle}}, \tag{32}$$

where $t_{i,\max}^{\mathrm{handle}}$ is the maximal acceptable handling latency.

### A.2.4 OBJECTIVE FORMULATION

In addition to the parcel handling latency, the robots' energy consumption is another important metric due to battery-powered characteristic. We need to constrain the energy budget. Let $v_i(t)$ (Joules) denote the budget of the robot $i$ at the time slot $t$ that can be spent on parcel handling. The average cumulative budget $\Upsilon_i$ (Joules) is defined to restrict the energy consumption of robot $i$:

$$E_i^f(t) = \varphi(v_{i,t}), \tag{33}$$

where $\varphi(x)$ is a mapping function with nonlinear character; $E_i^f(t)$ is a variable due to the undetermined obstacle distributions. Based on this, we can formulate the optimization objective as

$$P1 : \min \left\{ \lim_{T \to \infty} \frac{1}{T} \sum_{t=0}^{T} \Big[ \sum_{i=1}^{M} \sum_{k=1}^{K} a_1 \Delta T_k - a_2 \Delta A_{\mathrm{mix}} \Big] \right\},$$

$$\mathrm{s.t.} \left\{ \begin{array}{l} C1 : equation\ 17, equation\ 19, equation\ 25, equation\ 27, equation\ 32 \\ C2 : r_i \geq r_{\min}, \end{array} \right.$$

where $a_1$ and $a_2$ are weights in Lyapunov theory Matrosov (1962); $\Delta$ is the difference between the actual and virtual backlog. $\Delta T_k = b_{1,k} - b_{2,k}$, where $b_{1,k}$ and $b_{2,k}$ are the real handling latency and the expected handling latency for parcel $k$. We expect to minimize the difference value for real-time parcel handling. In terms of $C1$, equation 17, equation 19, equation 25, equation 27, and equation 32 ensure a comprehensive sensing cooperation, accurate parcel sorting with different delivery regions and destination, respectively, path conflict avoidance, and guarantee of low parcel handling latency. $C2$ can guarantee low-latency data transmission for real-time information exchanges among robots, cooperative FM training, and path planning computing.

### A.3 PSEUDOCODE

### A.4 DETAILED EXPERIMENT RESULTS

Fig. 8 illustrates the training loss of the cooperative FM under different numbers of robots, ranging from 20 to 60. We see that across all robots' numbers, the training loss decreases rapidly before iteration 200. The performance reflects efficient convergence at the initial stage. As training progresses, the loss gradually flattens and stabilizes, indicating that the model has reached convergence. In addition, we observe that configurations with more robots achieve lower training losses compared to scenarios with fewer robots. This trend demonstrates that increasing the number of cooperating

---

**Algorithm 1** Accurate parcel sorting and real-time handling with robust FMs.

---

**// Definition:** $\gamma = 0.99$.

**Input:** Network parameters $\theta$, state space $S_i$, action space $A_i$, update weight $\gamma$; replay buffer; system parameter set.

**Output:** Cooperative sorting and handling results.

 1: **Construct the FM cooperatively**
 2: **for** each episode in all the rounds **do**
 3:     Design the reward function using equation 2
 4:     **for** each time slot $t$ **do**
 5:         **for** each agent $i \in \mathcal{M}$ **do**
 6:             Obtain the feasible FM models by minimizing the $L$ using equation 3
 7:             Implement the customized sorting and handling models using equation 4
 8:             Obtain customized sub-FMs using 5 and 6
 9:         **end for**
10:     **end for**
11: **end for**
12: **Accurate parcel sorting**
13: Formulate feature vector $h_k$ using equation 7
14: **for** each parcel requirement **do**
15:     Acquire attention heads using equation 8
16:     Obtain the output of sub-FMs using equation 9
17: **end for**
18: **Real-time parcel handling**
19: **for** each iteration **do**
20:     Calculate the hidden state of GRU using equation 10
21:     Obtain the prediction results using equation 11
22: **end for**
23: **for** each exploration time **do**
24:     Update the positions and velocities using equation 12 and equation 13
25:     Update the global and local optimization results using equation 14
26: **end for**

---

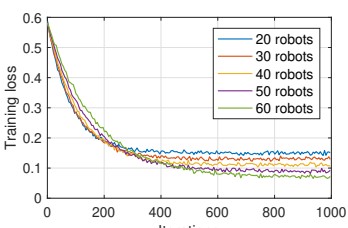

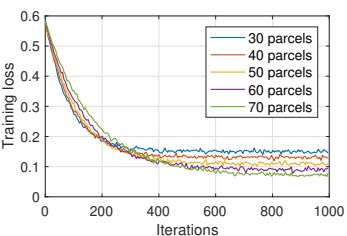

Figure 8: Training loss of cooperative FM with different numbers of robots.

Figure 9: Training loss of cooperative FM with different numbers of parcels.

robots enhances the representational capacity and leads to improved convergence performance. The results imply that our solution can achieve satisfied FM acquisition for efficient parcel sorting and handling.

Fig. 9 presents the training loss of the cooperative LAM model under different numbers of parcels. We see that the training loss for all solutions decreases sharply within the first 200 iterations followed by a gradual reduction and eventual stabilization. This behavior indicates that the model converges effectively under all workload conditions. Another important observation is that scenarios with a larger number of parcels attain lower training losses compared to cases with fewer parcels. This result demonstrates that increasing the task load can enhance final performance accuracy for accurate parcel sorting. Nevertheless, the convergence becomes slightly slower with higher parcel volumes, which reflects the trade-off between accuracy improvement and training efficiency. We can control the relations between the two indicators for high-efficiency sorting and handling.

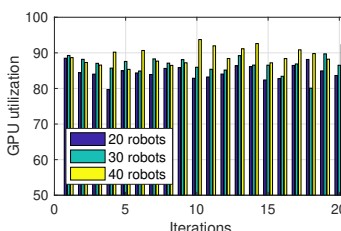

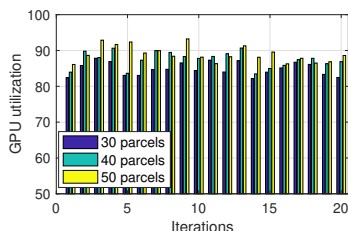

Figure 10: GPU utilization with different numbers of robots.

Figure 11: GPU utilization with different numbers of parcels.

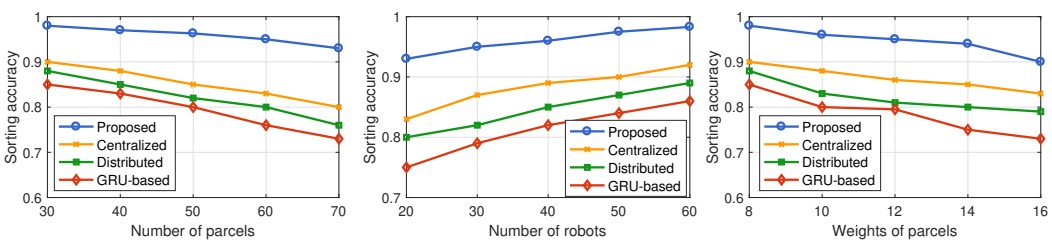

(a) Parcel sorting accuracy vs. number of parcels.

(b) Parcel sorting accuracy vs. number of robots.

(c) Parcel sorting accuracy vs. weights of parcels.

Figure 12: Performance evaluation of sorting accuracy under different numbers of robots, parcels, and weights of parcels.

Fig. 10 and Fig. 11 depict the GPU utilization of the cooperative FM framework under varying numbers of robots and parcels, respectively. In Fig. 10, the GPU utilization is compared across scenarios with 20, 30, and 40 robots. The results indicate that increasing the number of robots generally leads to higher and more stable GPU utilization. This suggests that larger cooperative groups are able to better exploit computational resources to minimize idle time and ensuring efficient parallel implementation with low execution latency. We can obtain the similar result in Fig. 11. We see that as the number of parcels increases, GPU utilization improves, reflecting the growing workload and enhanced resource occupancy. The higher task load ensures that the GPU remains consistently engaged, which improves training efficiency.

Fig. 5(c) illustrates the confidence error of FM training with respect to the number of iterations. The confidence error quantifies the gap between the predicted probability distribution and the ground truth, thereby reflecting the reliability of the decision-making. We see that all the solutions exhibit a decreasing confidence error with increasing iterations, indicating improved training stability. However, our solution consistently achieves lower confidence errors compared to other benchmarks. This demonstrates its ability to converge faster and maintain more reliable training predictions. Overall, the results highlight that our solution enhances both the convergence speed and the robustness of FM training, leading to safer and more reliable robot cooperation in parcel sorting and handling.

We compare parcel sorting accuracy in Fig. 12(a) under different numbers and types of parcels. With the random deployment of 40 robots with an average weight of 12 kg, our solution achieves the highest sorting accuracy compared to both benchmarks. This is due to our FM assists O-RAN in scheduling communication resources to ensure seamless cooperation among robots for efficient parcel sorting. Furthermore, our solution decouples missions into three sorting steps using our model-splitting method, which guarantees accurate parcel sorting through streamlined pipeline operations. The latency-FM-rApp also enables robots to sort parcels in real time for low latency requirements. Our solution improves the sorting accuracy by 7.1%, 13.8%, and 15% compared to centralized, distributed and GRU-based solutions, respectively.

Fig. 12(b) shows the performance of sorting accuracy under different numbers of robots. Given the 40 parcels with an average weight of 12 kg, our solution guarantees the highest sorting accuracy compared to all the benchmarks. This is because our FM can achieve deep collaboration among the

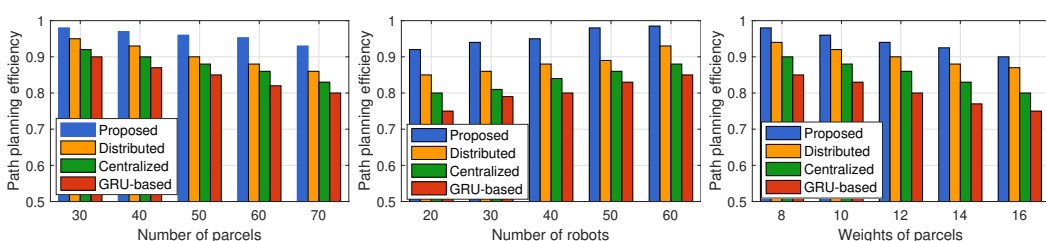

(a) Path planning efficiency vs. number of parcels.

(b) Path planning efficiency vs. number of robots.

(c) Path planning efficiency vs. weights of parcels.

Figure 13: Performance evaluation of path planning efficiency under different numbers of robots, parcels, and weights of parcels.

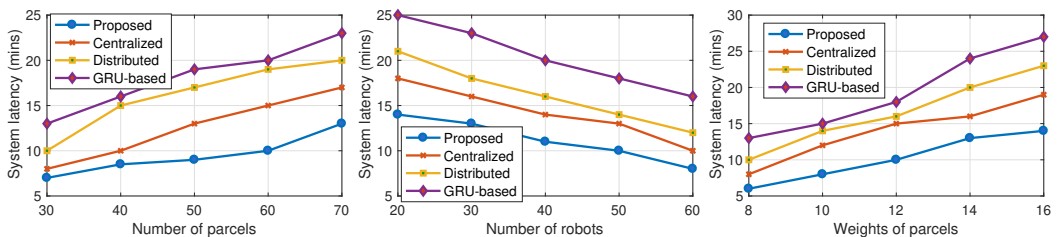

(a) System latency vs. number of parcels.

(b) System latency vs. number of robots.

(c) System latency vs. weights of parcels.

Figure 14: Performance evaluation of system latency under different numbers of robots, parcels, and weights of parcels.

R-FM-rApp, the D-FM-rApp, and L-FM-rApp to ensure accurate parcel sorting. In addition, our solution can collaborate computing resources of robots and edge RAN to improve the parcel sorting reliability for accurate parcel sorting. Furthermore, the latency-FM-rApp can conduct robots to optimize the sorting results considering different delivery priorities. Overall, our solution improves the sorting accuracy by 8%, 12.5%, and 15.9% compared to the centralized, distributed, and GRU-based algorithm, respectively.

We also illustrate the performance of sorting accuracy with different weights of parcels in Fig. 12(c). With 40 robots sorting 40 parcels, we can see that sorting accuracy reduces as the weights of parcels increase for all the solutions. This is because robots might cause a high recognition error for parcels with heavy weights due to large sizes of parcels. However, our solution still maintains a high sorting accuracy with up to 90% through robot cooperation. It is because our solution can assist robots in selecting feasible cooperators to implement cooperative sorting for high sorting accuracy. Additionally, our solution can collaborate feasible numbers of robots to match suitable numbers of parcels considering different weights of parcels. Overall, our solution improves the sorting accuracy by 9.2%, 15.9%, and 18.8% compared to the centralized, distributed, and GRU-based algorithm, respectively.

With the sorting results, we evaluate the efficiency of path planning, the ratio of the optimal path to the actual path with varying numbers of parcels, as illustrated in Fig. 13(a). Under the same deployment conditions as those in Fig. 12(a), our solution consistently achieves high path planning efficiency of up to 90% compared to both benchmarks. This is because our energy-FM-xApp allows robots to select feasible numbers of collaborators by estimating the energy consumption for cooperative parcel handling. Based on these estimation results, the PP-FM-xApp collaborates with the robots to implement cooperative path planning through information exchanges utilizing available communication resources. Our solution shortens the handling paths by 13.9% and 28.2% compared to distributed and centralized solutions, respectively.

With the same deployment as Fig. 12(b), Fig. 13(b) provides the path planning efficiency performance with different numbers of robots. The results clearly show that our solution consistently outperforms the other approaches across all robot configurations. In particular, our solution achieves efficiency levels close to or above 90%, even as the number of robots increases with high robustness and scalability. Another important observation is that increasing the number of robots slightly decreases the efficiency for all methods, reflecting the higher complexity of coordination in larger robot groups. Nevertheless, our method maintains a significant advantage over competing approaches, confirming its effectiveness for high-efficiency parcel handling. Our solution improve the path planning efficiency by 6.8%, 10.6%, and 17.5% compared to distributed, centralized and GRU-based solutions, respectively.

Under the same configuration as Fig. 12(c), we give the path planning efficiency performance with different weights of parcels in Fig. 13(c). It can be observed that our method consistently achieves the highest efficiency across all weight levels with up to 90% when the parcel weight is relatively low. Although efficiency decreases slightly as the parcel weight increases to 16, our solution still outperforms all alternatives by a significant margin. This also demonstrates that heavier parcels impose greater coordination challenges for all algorithms, but our method remains the most robust and scalable solution with the aid of terminal-edge cooperation. Our solution improve the path planning efficiency by 3.3%, 6.8%, and 17.5% compared to distributed, centralized and GRU-based solutions, respectively.

Finally, we compare system implementation latency under the different numbers of parcels shown in Fig. 14(a). With the same deployment scheme as shown in Fig. 12(a), we find that our solution consistently achieves the lowest implementation latency compared to both benchmarks. This improvement is attributed to our DDPG algorithm, which reduces the FM implementation time through a cooperative training manner by exchanging implementation actions. Additionally, our solution enables the O-RAN to collaborate with the computing resources of different robots to further accelerate FM training. We can dynamically schedule O-RAN communication resources to ensure reliable information exchanges for cooperative parcel sorting and handling. Our solution reduces system implementation latency by 15.8% and 30.4% compared to the distributed and centralized solutions, respectively.

With the same deployment as Fig. 12(b), Fig. 14(b) provides the system latency performance with different numbers of robots. We see that system latency decreases as the number of robots increases for all solutions, since more robots can share the workload and thus accelerate handling execution. However, our solution consistently achieves the lowest latency across all robot configurations, demonstrating superior scalability and efficiency. For example, with 60 robots, our solution reduces latency to nearly 5 minutes, significantly outperforming the centralized and distributed approaches, which remain above 10 minutes, and the GRU-based method, which exceeds 15 minutes. These results confirm that our solution not only leverages additional robotic resources more effectively but also minimizes communication and cooperation overhead.

Under the same configuration as Fig. 12(c), we give the system latency performance with different weights of parcels in Fig. 14(c). The results show that system latency increases with parcel weight for all solutions. This reflects the additional computational and cooperation overhead required for handling heavier parcels. Our solution consistently achieves the lowest latency across all weight values, maintaining latency below 15 minutes even for the heaviest parcels. These findings highlight that while heavier parcels impose greater system latency, our solution demonstrates superior robustness and scalability, effectively mitigating the latency increase compared to existing baselines. Our solution reduces system implementation latency by 33.3%, 41.2%, and 47.4% compared to the distributed, centralized, and GRU-based solutions, respectively.

