^{\text{R}} = \text{softmax}(W_{\text{R}}h_k), \quad p^{\text{D}} = \text{softmax}(W_{\text{D}}h_k), \quad p^{\text{L}} = \text{softmax}(W_{\text{L}}h_k), \tag{27}$$

where $W_{\text{R}}$, $W_{\text{D}}$, and $W_{\text{L}}$ are weight matrices. We employ the decoder to connect these attention heads with model parameters using a connection probability:

$$\mathcal{L}_{\text{R}-\text{FM}-\text{rApp}} = \alpha\big(-\log p_{y^{\text{R}}}^{\text{R}}\big), \quad \mathcal{L}_{\text{D}-\text{FM}-\text{rApp}} = \beta\big(-\log p_{y^{\text{D}}}^{\text{D}}\big), \quad \mathcal{L}_{\text{L}-\text{FM}-\text{rApp}} = \gamma\sum_k C_{y^{\text{L}},k}\, p_k^{\text{L}}, \tag{28}$$

where $\alpha$, $\beta$, and $\gamma$ are weights controlling the contribution of the region classification loss. $p^{\text{R}}$, $p^{\text{D}}$, and $p_k^{\text{L}}$ are probability distributions over all region classes; $y^{\text{R}}$ and $y^{\text{D}}$ are ground-truth class labels for the current samples. $C_{y^{\text{L}},k}$ is the penalty of predicting class $k$ when the true class is $y^{\text{L}}$. The latency-FM-rApp can assign appropriate robots to handle parcels based on distribution latency. Overall, the attention-based approach can reduce the time of model splitting while enhancing the accuracy of FM model implementation. We can transmit the sub-FMs to the Non-RT RICs to acquire various

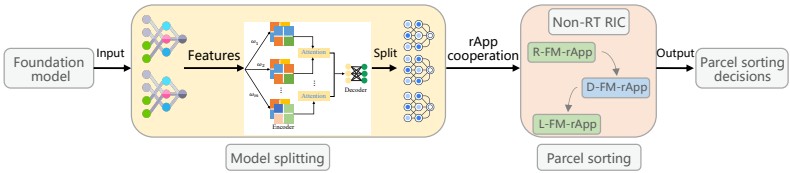

Figure 2: Illustration of parcel sorting.

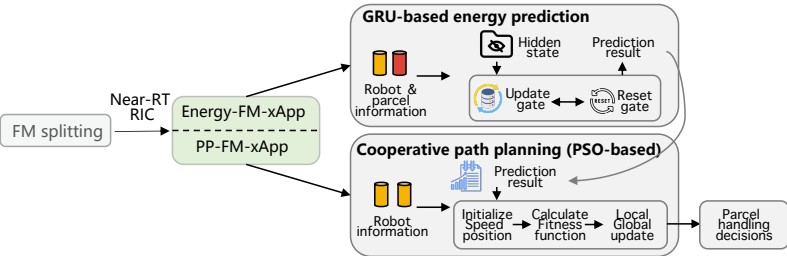

Figure 3: Illustration of cooperative parcel handling with two kinds of xApps, energy-FM-xApp (E-FM-xApp) and Path Planning (PP-FM-xApp).

rApps. This approach ensures accurate parcel sorting through deep collaboration throughout the entire pipeline. The algorithm is shown in Algorithm 1.

In addition, our pipeline approach can effectively adapt to changes in distribution requirements. When the destination requirements change, we can enable the Non-RT RIC to use the current FM-rApps to generate new sorting decisions by re-estimating distribution destinations for accurate parcel sorting. However, when the latency requirements change, the current FM-rApps may not be able to provide real-time parcel sorting due to their limited FM generation capabilities. To address this, we can implement a re-training operation to update the FM-rApps using an attention mechanism. This method can significantly reduce implementation latency by leveraging historical experiences.

### B.3 MODEL COOPERATION FOR REAL-TIME PARCEL HANDLING

We propose a cooperative parcel handling algorithm based on information on robots and parcels with two main implementation steps shown in Fig. 3: Gated Recurrent Unit (GRU)-based energy prediction and Particle Swarm Optimization (PSO)-based cooperative path planning.

In the Near-RT RIC, we input the robot and parcel information into the input gate. We then activate the reset gate $r_t$ using $\sigma$ function to determine how much of the previous hidden state should be forgotten when calculating the new hidden state:

$$r_t = \sigma(W_r x_{\text{en},t} + U_r h_{t-1} + b_r), \tag{29}$$

where $W_r$, $x_{\text{en},t}$, $U_r$, $h_{t-1}$ are weight matrix, input feature vector at time $t$, weight matrix for hidden state $h_t$, and the previous hidden state, respectively. $b_r$ is the bias vector. We can train the GRU network to acquire the accurate energy prediction result:

$$\mathcal{L}_{\text{pre}} = \frac{1}{T} \sum_{t=1}^{T} \left(\hat{E}_t - E_t\right)^2, \tag{30}$$

where $\hat{E}_t = W_y h_t + b_y$. With the prediction results, for robot $i$ at the dimension $d$, the position and velocity are updated as

$$V_{i,d} = \epsilon V_{i,d} + a_1 r_1 (X_d^{\text{g}} - X_{i,d}) + a_2 r_2 (X_d^{\text{p}} - X_{i,d}), \tag{31}$$

$$X_{i,d} = X_{i,d} + V_{i,d}, \tag{32}$$

where $\epsilon$ is the inertia parameter, $a_1$ and $a_2$ are acceleration coefficients; $r_1$ and $r_2$ are constrained in $[0, 1]$; $X^{\text{g}}$ and $X^{\text{p}}$ are global and local optimization results that are updated by

$$X_d^{\text{g}} = \arg\min_{X_d} g(X_{i,d}) = \sum_{j=1}^{M} \left\| X_{i,d} - h_j \right\|_2, \quad X_d^{\text{p}} = \arg\min_{X_d} g^{'}(X_{i,d}) = \sum_{j=1}^{M} \left\| X_{i,d} - h_j^{'} \right\|_2, \tag{33}$$

where $h_j$ and $h_j^{'}$ are global and local prediction values, respectively. Consequently, we can implement real-time and accurate parcel handling decisions using our designed path planning algorithm with the aid of Near-RT RIC. The whole algorithm implementation is shown in Algorithm 1.

Our solution enables a cooperative multi-task implementation by collaborating with rApps and xApps. First, we design three types of rApps, area-FM-rApp, destination-FM-rApp, and latency-FM-rApp, to perform cooperative parcel sorting in a pipeline manner. The sorting results from the latency-FM-rApp can be transmitted to the Near-RT RIC via the AI interface for parcel handling. We propose that the energy-FM-xApp and PP-FM-xApp implement cooperative path planning to achieve low-latency parcel handling with energy savings. In turn, the parcel handling results can be transmitted to the Non-RT RIC to assist the rApps in optimizing current computing resource scheduling decisions. Consequently, we can ensure high-efficiency parcel sorting and handling through deep cooperation between the Non-RT RIC and Near-RT RIC in robot-based mobile scenarios.

---