# OpenReview forum: "Robust Foundation Models Empowered RAN Intelligence for Reliable Embodied Robot Scenarios"
_ICLR.cc/2026/Conference — ICLR 2026 Conference Withdrawn Submission_

### Official Review · Reviewer_WMeP · 2025-10-20

**Soundness:** 3
**Presentation:** 2
**Contribution:** 2
**Rating:** 6
**Confidence:** 4

**Summary:**

This paper proposes a robust Foundation Model (FM)–empowered Open RAN (O-RAN) framework to enhance cooperation among autonomous warehouse robots performing parcel sorting and handling tasks. The framework integrates hierarchical FM training, adaptive model splitting, and cooperative path planning into the O-RAN control hierarchy, leveraging both Non-Real-Time (rApps) and Near-Real-Time (xApps) controllers.

The authors design two major algorithms:
An adaptive FM-splitting method that decomposes large models into region-, destination-, and latency-specific sub-FMs to improve sorting accuracy.
A cooperative handling algorithm combining GRU-based energy prediction and PSO-based path planning for real-time efficiency.

Experimental evaluation within a Simio-based 3 km × 3 km digital twin environment demonstrates up to 90 % sorting accuracy, a 13.9 % reduction in latency, and improved GPU utilization. The study benchmarks the system against centralized, distributed, and GRU-based baselines, reporting consistent performance improvements under various robot-parcel configurations.

**Strengths:**

*Timely integration of two research frontiers:* Combining Foundation Models with O-RAN for robotics is a novel and practically relevant direction.

*Hierarchical design clarity:* Clear separation between Non-RT and Near-RT RIC functions (rApps and xApps) improves interpretability and implementation feasibility.

*Algorithmic diversity:* Integrates reinforcement learning (DDPG), adversarially robust training, and swarm optimization, making the framework technically rich.

*Strong experimental grounding:* Employs realistic simulation tools, diverse metrics (accuracy, latency, GPU utilization, robustness), and multiple baselines.

*Security-aware training:* Inclusion of adversarial training and differential privacy in FM construction is commendable and often missing in RAN–robotics research.

*Reproducibility and detail:* Parameter settings, datasets (e.g., BubbleRAN, Kaggle warehouse management), and architecture breakdowns are explicitly provided.

**Weaknesses:**

Conceptual breadth vs. depth: The framework ambitiously merges FM design, RAN scheduling, and robotics control, but the paper lacks a unifying theoretical backbone connecting these domains.

Heavy reliance on simulation: No real-world robotic deployment is demonstrated; performance is shown only in digital twin environments.

*Evaluation against limited baselines:* The benchmarks do not include modern embodied-AI systems or edge-optimized foundation models such as RT-2, PaLM-E, or recent LLM-based control frameworks.

*Unclear computational cost trade-off:* Although GPU utilization and latency are measured, the absolute cost of FM training and inference on edge hardware is not quantified.

*Marginal novelty in individual components:* Techniques like GRU energy estimation or PSO path planning are standard; their integration is practical but not conceptually novel.

*Writing density and repetition:* Some sections (e.g., FM splitting and algorithm derivations) are verbose and could be condensed for readability.

**Questions:**

Detailed Analyses:
This work reflects the early confluence of foundation-level cognition and network-level intelligence. It captures a crucial shift in robotics — from standalone smart agents to collective intelligence orchestrated through communication infrastructure.

The study’s strength lies in its systems-level vision: robots are treated not as isolated learners but as federated nodes in a learning network — an O-RAN of cognition. That’s a profound perspective.

Yet, it also highlights a deeper tension: robustness vs. reality. The proposed adversarially trained FMs and dynamic RIC-based control appear robust in simulation, but robustness in communication reliability, model drift, and real-world packet loss is not yet demonstrated. The framework is impressive as a blueprint but remains one step away from proving operational maturity.

Still, the contribution is valuable because it documents a prototype paradigm — a modular, security-aware FM pipeline for edge robots — that future industrial systems can build upon.
This paper should be accepted conditionally, acknowledging its integrative ambition and engineering completeness, while recognizing the need for:

1. real-robot validation,
2. comparison with modern embodied FMs, and
3. clearer theoretical interpretation of FM-RAN coupling.

It is a strong systems paper rather than a theory paper, and within that boundary, it provides genuine novelty and utility to the robotics and networking communities. I expect the authors to defend or rebut the points in the weakness section during the rebuttal phase.

---

### Official Review · Reviewer_Nt2A · 2025-10-23

**Soundness:** 1
**Presentation:** 1
**Contribution:** 2
**Rating:** 2
**Confidence:** 2

**Summary:**

The paper proposes a distributed training algorithm for a transformer that allocates resources across a fleet of robots in a warehouse scenario. The method pretrains a transformer using DDPG. Subsequently, the model is finetuned. After training, a meta-learning algorithm is used to learn model splits for distributed inference. Given this training pipeline, the model is used for resource allocation, ie to say which robot should pick up which parcel, while a GRU-based method is used to learn energy expenditure of each robot for path planning which is performed using particle-swarm optimization.

**Strengths:**

- The model splitting part looks interesting. That being said, there are too few details to properly understand it in my eyes.
- The robustness analysis results  in Table 2 look kind of promising.

**Weaknesses:**

### Method
- The method seems to be relatively complex and unconventional. Pretraining an FM using RL and then finetuning using some kind of meta learning is definitely non-standard. While this is not a weakness per-se, it would be nice to understand the reasoning behind each of the choices better. Ideally the method should also be compared to the more standard recipe of supervised pretraining + some kind of (un) supervised finetuning to understand how it compares to that.
- Too many details of the method are unclear to me to judge it properly, for instance how the RL part is trained exactly and when Eq. 1 is optimized? I elaborate more on this below, but overall this makes it very difficult to judge the method.

### Evaluation:
- Some details are missing: do you report mean performance metrics over multiple training runs or what are the numbers in the tables?
- The training loss metric is difficult to interpret. Do you have some kind of validation performance over the course of training that you could report?
- Why are the parcels of weight 12kg according to the text in line 409, but table 1 lists 8 or 16?
- I am not sure if I follow your conclusion about the efficiency of your implementation based on the analysis of GPU usage. If you use more GPU memory because you must perform inference for more robots, that is quite logical to me. However, this does not mean that your implementation is efficient or that "larger cooperative groups are able to better exploit computational resources". Small groups just need less resources?

### Wording
- The categories listen in "Primary area" (alignment, fairness, safety, privacy, and societal considerations) have nothing to do with the paper it seems to me?
- I am not sure if the model that you train qualifies as a proper foundation model or if you simply train a transformer somehow. But I also must admit that the lines on this terrain are pretty blurred.

### Clarity:
- The paper is using a lot of abbreviations, in particular in the introduction. While some of them seem to be necessary, others are reused <=1 times, and in those cases it might aid reading flow to remove the abbreviation. Examples are Non-RT RIC, Near-RT RIC. Others are used but undefined, eg rApps in line 46, SMO in line 96, RU in Fig 1, and even using Google I cannot identify the meaning of all of them with certainty. Generally the paper is pretty hard to read for me due to the excessive use of abbreviations.
- Some parts of the method are not very clear to me. I think in some areas this might be related to the abbreviations and style of the writing, but in other parts information is clearly missing, see below. This must be fixed for future iterations of the paper.
- "Specifically, our algorithm can dynamically split the FM into three customized sub-FMs based on parcel information in the Non-RT RIC (at the SMO): Region-FM-rApp (R-FM-rApp), Destination-FM-rApp (D-FM-rApp), and Latency-FM-rApp (L-FM-rApp). The R-FM-rApp classifies parcels according to distribution regions" What do the D and L FM-rApps do?
- The model training pipeline is not entirely clear to me. Very fundamental aspects are not straightforward. For instance, if you train your model using supervised training, as suggested in eq. (1), where does the critic come from for your training?  In your text for Sec. 3.1. you should try to explain the training pipeline chronologically, ie, if the first stage of training is pre-training, start by defining the pretraining stage and ist losses / rewards, then proceed to explain the finetuning. Before going into the details, consider outlining the entire pipeline once in words.
- The text contains quite a few spelling and grammar errors. None of them are bad individually, but in their sum, it doesnt make the paper a good read and they should be fixed.
- "Based on this, we implement the fine-tuning training based on the parcel sorting and handling dataset using a Euclidean distance based meta-learning method" It is not clear to the reader what this means. You must explain this further in the paper.
- Some parts of the paper contain excessive information, eg Eq (8) or the explanation of a GRU should be clear to ICLR readers. Other parts of the method are then not explained at all which make it very difficult to follow.
- If you pretrain your model using RL, and you use DDPG, why do you choose this algorithm. Also state this crucial information in the method section and not only hidden in the evaluation section. Also DDPG is usually not databased and not directly an offline RL method? At the same time you state you use data for training, how do you do this?
- The color choices in Fig 7 are confusing. Dont swap colors for centralized and distributed between a and b, why do you introduce new colors in c?

### Minor:
- "The handling Path optimization is estimated with two key parts" path should not be capitalized here.
- "Robust FM construction FM splitting for accurate parcel sorting and FM cooperation for real-time parcel handling" There seems to be a comma missing.
- "where \theta is model parameters ( weights of a neural network)" There is an excessive white space and \theta *are* model parameters.
- "θ (x) is model prediction for input x under parameters θl L(·, ·) is cross-entropy measuring prediction error" The 1 here seems to be some typo?
- Perhaps opt for CE instead of L for the loss in (1) so it is directly clear which loss you optimize.
- Sometimes you state FM model, which is redundant. Use FM only or foundation model.
- "Based on the pre-training FM model, A data sampling operation can be enabled to decompose the data for customized parcel sorting and handling". Capitalization is off in this sentence.
- Figure 6 is not very interesting and should be removed.
- Table 1 should go into the appendix.
- The Adam citation in line 300 is off.
- "We see that across all robots’ numbers" should be "all robot numbers"

**Questions:**

- How and when do you finetune the model? I am not sure what exactly the meta-learning does here.
- Is the model splitting really leading to different parts of the model being on different devices? How can you dynamically change this if so?
- where do you use the PSO in Sec 3.3?
- Is it a valid assumption that inertia is only a scalar and not a matrix?
- Why do you use global and local optimization results in Eq 12?
- Why exactly do you have to split your model in the first place? Some of these algorithm choices seem to be hardly connected to the motivation in the introduction.
- In the training plots for the models, are iterations actually epochs, ie iterations over the entire dataset?
- Why do you plot different x axes in Fig 5? How do the curves in 5c evolve over the full 1000 iterations?
- Some of the results are not clear to me, for instance why exactly is your method lower latency if normally centralized beats distributed architectures by quite a bit? Your text explanation is confusing to me, why is training reducing implementation time and ow does this affect latency for inference?
- What is \Delta D in Eq 2? Why should \beta_i \in \{5, 0.05  \}?
- Is there any particular reason why you use PSO for path planning? Other evolutionary methods like CEM are much more standard in the robotics community.

---

### Official Review · Reviewer_tgKR · 2025-10-31

**Soundness:** 2
**Presentation:** 2
**Contribution:** 2
**Rating:** 4
**Confidence:** 2

**Summary:**

The paper proposes using a foundation model to coordinate warehouse robots through a wireless communication control framework. The model is divided into smaller modules that operate at different time scales: higher-level modules (“rApps”) perform parcel-sorting decisions, while lower-level modules (“xApps”) coordinate robot movement, energy use, and path planning. The system also includes a GRU-based energy predictor, a particle swarm optimizer for motion coordination, and robustness mechanisms such as adversarial training and differential privacy.

The idea is interesting, bridging large-scale models with multi-robot coordination. However, the paper assumes substantial background knowledge of O-RAN, appears to make overstated novelty claims, and lacks clear evidence supporting some of its key assertions, particularly around the supposed difficulty of the task and the meaning of its reported performance metrics.

**Strengths:**

- Integration of communication and learning: The idea of linking foundation models with wireless coordination aligns well with current trends in edge intelligence.
- Hierarchical design: The split between rApps and xApps is conceptually consistent with O-RAN’s Non-Real-Time and Near-Real-Time control structure.
- Attention to robustness: The inclusion of adversarial training and differential privacy is a positive step toward trustworthy system design.
- Comprehensive performance metrics: The authors attempt to measure multiple factors, including latency, sorting accuracy, energy, and reliability.

**Weaknesses:**

**Requires O-RAN background knowledge**

The paper heavily relies on O-RAN terminology (rApps, xApps, RIC, SMO) without defining these terms. Readers unfamiliar with this architecture will find it difficult to understand how the system components interact or what makes the proposed framework distinct. A short, self-contained description of the O-RAN layers and their time scales would make the paper far more approachable.

**Novelty claims are overstated or not properly put in context**

The authors describe this as the "first exploration" of integrating foundation models with O-RAN for robotics. However, they also list related works that appear to combine large models and O-RAN-like systems—for example:
- Tang et al. (2023): AI testing framework for next-generation O-RAN networks.
- Xia et al. (2024): agent-based foundation models for collaborative vehicles.
- Sun et al. (2024) and Niyato et al. (2024): generative AI for dynamic networking.

The contribution here appears to be an application of existing AI-RAN and FM-networking concepts to warehouse robotics, not a fundamentally new paradigm.

**Insufficient support for claimed "critical challenges'"**

In the introduction, the paper asserts that parcel sorting faces critical challenges due to limited onboard compute, constrained communication resources, and cloud latency. However, these claims are not backed by quantitative evidence, detailed discussion, or supporting work.

From my perspective, it is unclear why this particular problem is so difficult or why the system achieves only around 80% sorting accuracy in the non-adversarial case. Is this due to noisy perception, communication bottlenecks, task coordination complexity, or other limitations? Without explanation, it is hard to assess whether 80% reflects a genuine bottleneck or simply an unoptimized setup.

Providing an error analysis or baseline comparison (for example, single-robot or perfect-communication cases) would help justify the challenge and clarify what limits performance.

**Experimental transparency and rigor**

- The reported learning rate of 0.99 for DDPG training is almost certainly incorrect.
- Simulations rely on Simio and a "Bubble RAN"' environment, but no code, configurations, or dataset splits are released.
- Datasets are only referenced via Kaggle links without describing any preprocessing or labeling (or lack thereof)
- No ablations are provided to isolate the effects of FM-splitting, GRU-based energy modeling, or PSO path planning.

**Questions:**

1. The paper highlights several critical challenges (e.g. limited onboard computation, communication constraints, and latency) but does not clearly illustrate how these factors impact task performance in the parcel sorting task. It would be helpful to expand on why parcel sorting is difficult in this setting and what primarily limits system accuracy (e.g., perception noise, coordination complexity, timing delays, or other factors). Providing additional context or discussion here would help readers understand the underlying sources of difficulty and interpret the reported 80% accuracy more meaningfully.
2. How does your FM-splitting approach compare to simpler baselines, such as a single unsplit model or a system without O-RAN layering? Ablations isolating these factors would strengthen your claims of improvement.
3. The paper lists a DDPG learning rate of 0.99. Is this a typographical error, or does it refer to another parameter such as a discount factor or target-update rate?

---

### Official Review · Reviewer_vu81 · 2025-10-31

**Soundness:** 2
**Presentation:** 2
**Contribution:** 2
**Rating:** 2
**Confidence:** 2

**Summary:**

This paper proposes a framework to improve the robustness, accuracy, and efficiency of multi-robot cooperation for parcel sorting in a warehouse scenario. The core proposal is the integration of Foundation Models (FMs) with an Open Radio Access Network (O-RAN) architecture. The paper claims this is the first exploration of such an integration for robotic applications.

The key claims are:
Framework: A robust, FM-empowered O-RAN framework that uses a terminal-edge cooperative paradigm. The O-RAN's RICs (RAN Intelligent Controllers) are leveraged to manage tasks, with rApps in the Non-RT RIC and xApps in the Near-RT RIC.
Robust Training: A robust FM construction method (Section 3.1) based on min-max adversarial training (Eq. 1) and DDPG-based pre-training (Eq. 3) to defend against various security risks.
Sorting Algorithm: An "adaptive FM-splitting" algorithm (Section 3.2) deployed in the Non-RT RIC. This method uses an attention mechanism (Eq. 8) to create specialized sub-FMs (R-FM-rApp, D-FM-rApp, L-FM-rApp) to improve sorting accuracy.
Handling Algorithm: A "cooperative path planning" algorithm (Section 3.3) in the Near-RT RIC. This algorithm is composed of a GRU-based model for energy prediction (E-FM-xApp) and a PSO-based algorithm for path planning (PP-FM-xApp) to reduce latency.
Empirical Claims: Simulation results (Section 4) demonstrate that the proposed framework achieves high sorting accuracy (up to 90%, Fig 7a), high path planning efficiency (up to 90%, Fig 7b), and significantly lower system latency (Fig 7c) compared to centralized FM, distributed FM, and GRU-based baselines. The paper also claims competitive performance in adversarial robustness (Table 2).

**Strengths:**

Timely and Novel Problem: The paper addresses a significant and timely problem at the intersection of large-scale AI (Foundation Models), robotics, and next-generation communication networks (O-RAN). The application to warehouse logistics is practical and well-motivated.

System-Level Perspective: The authors propose a complete, hierarchical system architecture. The thoughtful integration of computation (FMs) with network orchestration (O-RAN RICs, rApps, xApps) and a multi-level task structure (Non-RT for sorting, Near-RT for handling) is a non-trivial system design contribution.

Comprehensive Problem Formulation: The work attempts to tackle multiple critical facets of the problem simultaneously: security/robustness (adversarial training), task accuracy (FM splitting for sorting), and real-time efficiency (path planning, energy estimation).

Extensive Simulation: The empirical evaluation in Section 4 is extensive, testing the system's performance across a variety of metrics (accuracy, latency, efficiency, training loss, GPU use) and parameters (number of robots, parcels, and parcel weights).

**Weaknesses:**

Critical Lack of Clarity in Methodology: The paper's primary weakness is that the core technical contributions are poorly explained to the point of being irreproducible. (i) What is the "Foundation Model"? The paper's central component, the FM, is never defined. The introduction mentions Transformers, but the methodology in Section 3.1 describes a "pre-training" process using DDPG (an RL algorithm, see Fig. 2, Eq. 3). Section 3.3 then introduces GRU and PSO. The actual architecture of the FM (e.g., model type, layers, parameters) is completely missing. This makes the title and central claim of "FM-empowered" intelligence questionable. (ii) "FM-Splitting" (Sec 3.2) is Unclear: The mechanism for "splitting" the FM is opaque. The paper presents an attention mechanism (Eq. 8) and loss functions (Eq. 9). It is not explained how this "splits" the model. Is this just multi-task learning with different heads? Are model parameters being partitioned? How are the "sub-FMs" (rApps) derived from the main FM?

Lacks Strong Connection Between Claims and Evidence: Robustness/Security (Tables 2 & 3): The empirical support for the robustness claims is weak. In Table 2, the proposed method ("Ours") shows only a marginal improvement over the "AdvTrain (PGD-10)" baseline. In Table 3 (Security evaluation), the proposed method is not even included as an entry, making it impossible to evaluate the security claims. The text's assertion that the solution "combines complementary approaches" is not demonstrated.

Lacks Evaluation Baselines: The main empirical results (Fig. 7, 12-14) rely on comparisons to "centralized FM" and "distributed FM" baselines. These baselines are not defined. What are their architectures? How are they trained? Were they also adversarially trained for a fair comparison? Without this information, the strong empirical claims (e.g., 13.9% latency reduction) are unsubstantiated.

**Questions:**

What is the specific neural network architecture (e.g., Transformer, MLP, etc.), including layer types and sizes, of the "Foundation Model" at the heart of your framework? How does the DDPG algorithm (Sec 3.1) serve as a "pre-training" method for this FM?

Please clarify the role of the FM in Section 3.3. Are the E-FM-xApp and PP-FM-xApp just a standard GRU and a standard PSO algorithm, respectively? If so, where is the FM, and what role does it play?

Please explain the complete training pipeline. How are the min-max adversarial loss (Eq. 1), the DDPG pre-training (Eq. 3), and the supervised losses (Eq. 5, 6) combined into a single, coherent training process?

What are the architectures of the "centralized FM" and "distributed FM" baselines used in the evaluation, and how were they trained?

---

### Note · Authors · 2025-11-28

I have read and agree with the venue's withdrawal policy on behalf of myself and my co-authors.